# Progressive Memory Banks for Incremental Domain Adaptation

**Nabiha Asghar**[*][†][◦] **Lili Mou**[*][‡] **Kira A. Selby**[†][◦] **Kevin D. Pantasdo**[◦] **Pascal Poupart**[†][◦] **Xin Jiang**[◇]

[†]Vector Institute for AI, Toronto, Canada

[◦]Cheriton School of Computer Science, University of Waterloo, Canada

`{nasghar,kaselby,kevin.pantasdo,ppoupart}@uwaterloo.ca`

[‡]Dept. Computing Science, University of Alberta; Alberta Machine Intelligence Institute (AMII)

`doublepower.mou@gmail.com`

[◇]Noah's Ark Lab, Huawei Technologies, Hong Kong

`jiang.xin@huawei.com`

## Abstract

This paper addresses the problem of incremental domain adaptation (IDA) in natural language processing (NLP). We assume each domain comes one after another, and that we could only access data in the current domain. The goal of IDA is to build a unified model performing well on all the domains that we have encountered. We adopt the recurrent neural network (RNN) widely used in NLP, but augment it with a directly parameterized memory bank, which is retrieved by an attention mechanism at each step of RNN transition. The memory bank provides a natural way of IDA: when adapting our model to a new domain, we progressively add new slots to the memory bank, which increases the number of parameters, and thus the model capacity. We learn the new memory slots and fine-tune existing parameters by back-propagation. Experimental results show that our approach achieves significantly better performance than fine-tuning alone. Compared with expanding hidden states, our approach is more robust for old domains, shown by both empirical and theoretical results. Our model also outperforms previous work of IDA including elastic weight consolidation and progressive neural networks in the experiments.[1]

## 1 Introduction

Domain adaptation aims to transfer knowledge from one domain (called the *source domain*) to another (called the *target domain*) in a machine learning system.[2] If the data of the target domain are not large enough, using data from the source domain typically helps to improve model performance in the target domain. This is important for neural networks, which are data-hungry and prone to overfitting. In this paper, we especially focus on *incremental domain adaptation* (IDA)[3], where we assume different domains come sequentially one after another. We only have access to the data in the current domain, but hope to build a unified model that performs well on all the domains that we have encountered (Xu et al., 2014; Rusu et al., 2016; Kirkpatrick et al., 2017).

Incremental domain adaptation is useful in scenarios where data are proprietary, or available only for a short period of time (Li & Hoiem, 2018). It is desired to preserve as much knowledge as possible in the learned model and not to rely on the availability of the data. Another application of IDA is a quick adaptation to new domains. If the environment of a deployed machine learning system changes frequently, traditional methods like jointly training all domains require the learning machine to be re-trained from scratch every time a new domain comes. Fine-tuning a neural network by a

---

[*] Equal contribution.

[1]Our IDA code is available at `https://github.com/nabihach/IDA`.

[2]In our work, the *domain* is defined by datasets. Usually, the data from different genres or times typically have different underlying distributions.

[3]In the literature, IDA is sometimes referred to as *incremental*, *continual*, or *lifelong learning*. We use "incremental domain adaptation" in this paper to emphasize that our domain is not changed continuously.

few steps of gradient updates does transfer quickly, but it suffers from the *catastrophic forgetting problem* (Kirkpatrick et al., 2017). Suppose during prediction a data point is not labeled with its domain, the (single) fine-tuned model cannot predict well for samples in previous domains, as it tends to "forget" quickly during fine-tuning.

A recent trend of domain adaptation in the deep learning regime is the progressive neural network (Rusu et al., 2016), which progressively grows the network capacity if a new domain comes. Typically, this is done by enlarging the model with new hidden states and a new predictor (Figure 1a). To avoid interfering with existing knowledge, the newly added hidden states are not fed back to the previously trained states. During training, all existing parameters are frozen, and only the newly added ones are trained. For inference, the new predictor is used for all domains, which is sometimes undesired as the new predictor is trained with only the last domain.

In this paper, we propose a progressive memory bank for incremental domain adaptation in natural language processing (NLP). Our model augments a recurrent neural network (RNN) with a memory bank, which is a set of distributed, real-valued vectors capturing domain knowledge. The memory is retrieved by an attention mechanism during RNN information processing. When our model is adapted to new domains, we progressively increase the slots in the memory bank. But different from Rusu et al. (2016), we fine-tune all the parameters, including RNN and the previous memory bank. Empirically, when the model capacity increases, the RNN does not forget much even if the entire network is fine-tuned. Compared with expanding RNN hidden states, the newly added memory slots cause less contamination of the existing knowledge in RNN states, as will be shown by a theorem.

In our paper, we evaluate the proposed approach on a classification task known as multi-genre natural language inference (MultiNLI). Appendix C provides additional evidence when our approach is applied to text generation. Experiments consistently support our hypothesis that the proposed approach adapts well to target domains without catastrophic forgetting of the source. Our model outperforms the naïve fine-tuning method, the original progressive neural network, as well as other IDA techniques including elastic weight consolidation (EWC, Kirkpatrick et al., 2017).

## 2 RELATED WORK

**Domain Adaptation.** Domain adaptation has been widely studied in machine learning, including the NLP domain. For neural NLP applications, Mou et al. (2016) analyze two straightforward settings, namely, multi-task learning (jointly training all domains) and fine-tuning (training one domain and fine-tuning on the other). A recent advance of domain adaptation is adversarial learning, where the neural features are trained not to classify the domain (Ganin et al., 2016; Liu et al., 2017). However, all these approaches (except fine-tuning) require all domains to be available simultaneously, and thus are not IDA approaches.

Kirkpatrick et al. (2017) address the catastrophic forgetting problem of fine-tuning neural networks, and propose a regularization term based on the Fisher information matrix; they call the method elastic weight consolidation (EWC). While some follow-up studies report EWC achieves high performance in their scenarios (Zenke et al., 2017; Lee et al., 2017; Thompson et al., 2019), others show that EWC is less effective (Wen & Itti, 2018; Yoon et al., 2018; Wu et al., 2018). Lee et al. (2017) propose incremental moment matching between the posteriors of the old model and the new model, achieving similar performance to EWC. Schwarz et al. (2018) augment EWC with knowledge distillation, making it more memory-efficient.

Rusu et al. (2016) propose a progressive neural network that progressively increases the number of hidden states (Figure 1a). To avoid overriding existing information, they freeze the weights of the learned network, and do not feed new states to old ones. This results in multiple predictors, requiring that a data sample is labeled with its domain during the test time. If we otherwise use the last predictor to predict samples from all domains, its performance may be low for previous domains, as the predictor is only trained with the last domain.

Yoon et al. (2018) propose an extension of the progressive network. They identify which existing hidden units are relevant for the new task (with their sparse penalty), and fine-tune only the corresponding subnetwork. However, sparsity is not common for RNNs in NLP applications, as sparse recurrent connections are harmful. A similar phenomenon is that dropout of recurrent connections

Figure 1: (a) Progressive neural network Rusu et al. (2016). (b) One step of RNN transition in our progressive memory network. Colors indicate different domains.

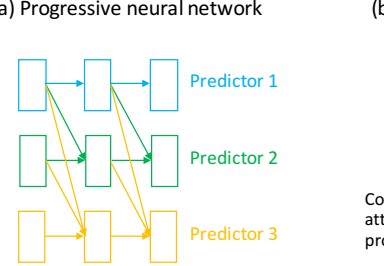
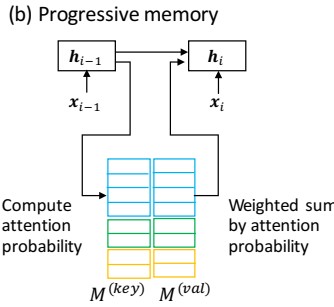

yields poor performance (Bayer et al., 2013). Xu & Zhu (2018) deal with new domains by adaptively adding nodes to the network via reinforcement learning. This approach may require a very large number of trials to identify the right number of nodes to be added to each layer (Yoon et al., 2019).

Li & Hoiem (2018) address IDA with a knowledge distillation approach, where they preserve a set of outputs of the old network on pseudo-training data. Then they jointly optimize for the accuracy on the new training domain as well as the pseudo-training data. Kim et al. (2019)'s variant of this approach uses maximum-entropy regularization to control the transfer of distilled knowledge. However, in NLP applications, it is non-trivial to obtain pseudo-training data for distillation.

**Memory-Based Neural Networks.** Our work is related to memory-based neural networks. Sukhbaatar et al. (2015) propose an end-to-end memory network that assigns each memory slot to an entity, and aggregates information by multiple attention-based layers. They design their architecture for the bAbI question answering task, and assign a slot to each sentence. Such idea can be extended to various scenarios, for example, assigning slots to external knowledge for question answering (Das et al., 2017) and assigning slots to dialogue history for a conversation system (Madotto et al., 2018).

A related idea is to use episodic memory, which stores data samples from all previously seen domains (thus it is not an IDA approach). This is used for experience replay while training on subsequent domains (Lopez-Paz & Ranzato, 2017; Rebuffi et al., 2017; Chaudhry et al., 2018; d'Autume et al., 2019).

Another type of memory in the neural network regime is the neural Turing machine (NTM, Graves et al., 2016). This memory is not directly parameterized, but is read or written by a neural controller. Therefore, it serves as temporary scratch paper and does not store knowledge itself. Zhang et al. (2018b) combine the above two styles of memory for task-oriented dialogue systems, where they have both slot-value memory and read-and-write memory.

Different from the work above, our memory bank stores knowledge in a distributed fashion, where each slot does not correspond to a concrete entity or data sample. Our memory is directly parameterized, interacting in a different way from RNN weights and providing a natural way of incremental domain adaptation.

## 3 PROPOSED APPROACH

Our model is based on a recurrent neural network (RNN). At each step, the RNN takes the embedding of the current input (e.g., a word), and changes its states accordingly. This is represented by $\boldsymbol{h}_i = \text{RNN}(\boldsymbol{h}_{i-1}, \boldsymbol{x}_i)$, where $\boldsymbol{h}_i$ and $\boldsymbol{h}_{i-1}$ are the hidden states at time steps $i$ and $i-1$, respectively. $\boldsymbol{x}_i$ is the input at the $i$th step. Typically, long short term memory (LSTM, Hochreiter & Schmidhuber, 1997) or Gated Recurrent Units (GRU, Cho et al., 2014) are used as RNN transitions. In the rest of this section, we will describe a memory-augmented RNN, and how it is used for incremental domain adaptation (IDA).

## 3.1 Augmenting RNN with Memory Banks

We enhance the RNN with an external memory bank, as shown in Figure 1b. The memory bank augments the overall model capacity by storing additional parameters in memory slots. At each time step, our model computes an attention probability to retrieve memory content, which is then fed to the computation of RNN transition.

Particularly, we adopt a key-value memory bank, inspired by Miller et al. (2016). Each memory slot contains a key vector and a value vector. The former is used to compute the attention weight for memory retrieval, whereas the latter is the value of memory content.

For the $i$th step, the memory mechanism computes an attention probability $\boldsymbol{\alpha}_i$ by

$$\widetilde{\alpha}_{i,j} = \exp\{\boldsymbol{h}_{i-1}^\top \boldsymbol{m}_j^{(\text{key})}\}, \quad \alpha_{i,j} = \frac{\widetilde{\alpha}_{i,j}}{\sum_{j'=1}^{N} \widetilde{\alpha}_{i,j'}} \tag{1}$$

where $\boldsymbol{m}_j^{(\text{key})}$ is the key vector of the $j$th slot of the memory (among $N$ slots in total). Then the model retrieves memory content by a weighted sum of all memory values, where the weight is the attention probability, given by

$$\boldsymbol{c}_i = \sum_{j=1}^{N} \alpha_{i,j} \boldsymbol{m}_j^{(\text{val})} \tag{2}$$

Here, $\boldsymbol{m}_j^{(\text{val})}$ is the value vector of the $j$th memory slot. We call $\boldsymbol{c}_i$ the *memory content*. Then, $\boldsymbol{c}_i$ is concatenated with the current word $\boldsymbol{x}_i$, and fed to the RNN as input for state transition.

Using the key-value memory bank allows separate (thus more flexible) computation of memory retrieval weights and memory content, compared with traditional attention where a candidate vector is used to compute both attention probability and attention content.

It should be emphasized that the memory bank in our model captures distributed knowledge, which is different from other work where the memory slots correspond to specific entities (Eric et al., 2017). The attention mechanism accomplishes memory retrieval in a "soft" manner, which means the retrieval strength is a real-valued probability. This enables us to train both memory content and its retrieval end-to-end, along with other neural parameters.

We would also like to point out that the memory bank alone does not help RNN much. However, it is natural to use a memory-augmented RNN for incremental domain adaptation, as described below.

## 3.2 Progressively Increasing Memory for Incremental Domain Adaptation

The memory bank in Subsection 3.1 can be progressively expanded to adapt a model in a source domain to new domains. This is done by adding new memory slots to the bank which are learned exclusively from the target data.

Suppose the memory bank is expanded with another $M$ slots in a new domain, in addition to previous $N$ slots. We then have $N + M$ slots in total. The model computes attention probability over the expanded memory and obtains the attention vector in the same way as Equations (1)–(2), except that the summation is computed from 1 to $N + M$. This is given by

$$\alpha_{i,j}^{(\text{expand})} = \frac{\widetilde{\alpha}_{i,j}}{\sum_{j'=1}^{N+M} \widetilde{\alpha}_{i,j'}}, \quad \boldsymbol{c}_i^{(\text{expand})} = \sum_{j=1}^{N+M} \alpha_{i,j}^{(\text{expand})} \boldsymbol{m}_j^{(\text{val})} \tag{3}$$

To initialize the expanded model, we load all previous parameters, including RNN weights and the learned $N$ slots, but randomly initialize the progressively expanded $M$ slots. During training, we update all parameters by gradient descent. That is to say, new parameters are learned from their initializations, whereas old parameters are fine-tuned during IDA. The process is applied whenever a new domain comes, as shown in Algorithm 1.

We would like to discuss the following issues.

**Freezing vs. Fine-tuning learned parameters.** Inspired by the progressive neural network (Rusu et al., 2016), we find it tempting to freeze RNN parameters and the learned memory but only tune new memory for IDA. However, our preliminary results show that if we freeze all existing parameters, the increased memory does not add much to the model capacity, and that its performance is worse than fine-tuning all parameters.

**Fine-tuning vs. Fine-tuning while increasing memory slots.** It is reported that fine-tuning a model (without increasing model capacity) suffers from the problem of catastrophic forgetting (Kirkpatrick et al., 2017). We wish to investigate whether our approach suffers from the same problem, since we fine-tune learned parameters when progressively increasing memory slots. Our intuition is that the increased model capacity helps to learn the new domain with less overriding of the previously learned model. Experiments confirm our conjecture, as the memory-augmented RNN tends to forget more if the memory size is not increased.

**Expanding hidden states vs. Expanding memory.** Another way of progressively increasing model capacity is to expand the size of RNN layers. This setting is similar to the progressive neural network, except that all weights are fine-tuned and new states are connected to existing states.

However, we hereby show a theorem, indicating that the expanded memory results in less contamination/overriding of the learned knowledge in the RNN, compared with the expanded hidden states. The main idea is to measure the effect of model expansion quantitatively by the expected square difference on $\boldsymbol{h}_i$ before and after expansion, where the expectation reflects the average effect of model expansion in different scenarios.

**Theorem 1.** *Let RNN have vanilla transition with the linear activation function, and let the RNN state at the last step $\boldsymbol{h}_{i-1}$ be fixed. For a particular data point, if the memory attention satisfies $\sum_{j=N+1}^{N+M} \widetilde{\alpha}_{i,j} \leq \sum_{j=1}^{N} \widetilde{\alpha}_{i,j}$, then memory expansion yields a lower expected mean squared difference in $\boldsymbol{h}_i$ than RNN state expansion. That is,*

$$\mathbb{E}\left[\|\boldsymbol{h}_i^{(\mathrm{m})} - \boldsymbol{h}_i\|^2\right] \leq \mathbb{E}\left[\|\boldsymbol{h}_i^{(\mathrm{s})} - \boldsymbol{h}_i\|^2\right] \tag{4}$$

*where $\boldsymbol{h}_i^{(\mathrm{m})}$ refers to the hidden states if the memory is expanded. $\boldsymbol{h}_i^{(\mathrm{s})}$ refers to the original dimensions of the RNN states, if we expand the size of RNN states themselves. Here, we compute the expectation by assuming weights and hidden states are iid from a zero-mean Gaussian distribution (with variance $\sigma^2$).*

**Proof Sketch:** We focus on one step of RNN transition and assume that $\boldsymbol{h}_{i-1}$ is the same when the model capacity is increased. Further, we assume that $\boldsymbol{h}_i$ is $D$-dimensional, that each memory slot $\boldsymbol{m}_j$ is $d$-dimensional, and that the additional RNN units (when we expand the hidden state) are also $d$-dimensional.

We compute how the original dimensions in the hidden state are changed if we expand RNN. We denote the expanded hidden states by $\widetilde{\boldsymbol{h}}_{i-1}$ and $\widetilde{\boldsymbol{h}}_i$ for the two time steps. We denote the weights connecting from $\widetilde{\boldsymbol{h}}_{i-1}$ to $\boldsymbol{h}_i$ by $\widetilde{W} \in \mathbb{R}^{D \times d}$. We focus on the original $D$-dimensional space, denoted as $\boldsymbol{h}_i^{(s)}$. The connection is shown in Figure 2a, Appendix A. We have

$$\mathbb{E}\left[\|\boldsymbol{h}_i^{(\mathrm{s})} - \boldsymbol{h}_i\|^2\right] = \mathbb{E}\left[\|\widetilde{W} \cdot \widetilde{\boldsymbol{h}}_{i-1}\|^2\right] = \sum_{j=1}^{D} \mathbb{E}\left[\left(\widetilde{\boldsymbol{w}}_j^\top \widetilde{\boldsymbol{h}}_{i-1}\right)^2\right] \tag{5}$$

$$= \sum_{j=1}^{D} \mathbb{E}\left[\left(\sum_{k=1}^{d} \widetilde{w}_{jk}\widetilde{h}_{i-1}[k]\right)^2\right] = \sum_{j=1}^{D}\sum_{i=1}^{d} \mathbb{E}\left[\left(\widetilde{w}_{jk}\right)^2\right]\mathbb{E}\left[\left(\widetilde{h}_{i-1}[k]\right)^2\right] \tag{6}$$

$$= D \cdot d \cdot \mathrm{Var}(w) \cdot \mathrm{Var}(h) = Dd\sigma^2\sigma^2 \tag{7}$$

---

**Algorithm 1:** Progressive Memory for IDA

**Input**: A sequence of domains $D_0, D_1, \cdots, D_n$
**Output**: A single model for all domains
Initialize a memory-augmented RNN
Train the model on $D_0$
**for** $D_1, \cdots, D_n$ **do**
    Expand the memory with new slots
    Load RNN weights and existing memory banks
    Train the model by updating all parameters
**end**
**Return**: The resulting model

Table 1: Corpus statistics and the baseline performance (% accuracy) of our BiLSTM model (without domain adaptation) and results reported in previous work.

| | Fic | Gov | Slate | Tel | Travel |
|---|---|---|---|---|---|
| # training samples | 77k | 77k | 77k | 83k | 77k |
| Our Implementation | 65.0 | 66.5 | 56.2 | 64.5 | 62.7 |
| Yu et al. (2018) | 64.7 | 69.2 | 57.9 | 64.4 | 65.8 |

Table 2: Results on two-domain adaptation. F: Fine-tuning. V: Expanding vocabulary. H: Expanding RNN hidden states. M: Our proposed method of expanding memory. We also compare with previous work elastic weight consolidation (EWC, Kirkpatrick et al., 2017) and the progressive neural network (Rusu et al., 2016). For the statistical test (compared with Line 8), $\uparrow, \downarrow$: $p < 0.05$ and $\Uparrow, \Downarrow$: $p < 0.01$. The absence of an arrow indicates that the performance difference compared with Line 8 is statistically insignificant with $p$ lower than 0.05.

| #Line | Model | Trained on/by | % Accuracy on | |
|---|---|---|---|---|
| | | | S | T |
| 1 | RNN | S | $65.01^{\Downarrow}$ | $61.23^{\Downarrow}$ |
| 2 | | T | $56.46^{\Downarrow}$ | $66.49^{\Downarrow}$ |
| 3 | RNN+ Mem | S | $65.41^{\Downarrow}$ | $60.87^{\Downarrow}$ |
| 4 | | T | $56.77^{\Downarrow}$ | $67.01^{\Downarrow}$ |
| 5 | | S+T | $66.02^{\downarrow}$ | 70.00 |
| 6 | RNN + Mem | S→T (F) | $65.62^{\downarrow}$ | $69.90^{\downarrow}$ |
| 7 | | S→T (F+M) | 66.23 | 70.21 |
| 8 | | S→T (F+M+V) | **67.55** | **70.82** |
| 9 | | S→T (F+H) | $64.09^{\Downarrow}$ | $68.35^{\Downarrow}$ |
| 10 | | S→T (F+H+V) | $63.68^{\Downarrow}$ | $68.02^{\Downarrow}$ |
| 11 | | S→T (EWC) | $66.02^{\Downarrow}$ | $64.10^{\Downarrow}$ |
| 12 | | S→T (Progressive) | $64.47^{\Downarrow}$ | $68.25^{\Downarrow}$ |

Similarly,

$$\mathbb{E}\big[\|\boldsymbol{h}_i^{(\mathrm{m})} - \boldsymbol{h}_i\|^2\big] = \mathbb{E}\Big[\big\|W_{(c)}\Delta\boldsymbol{c}\big\|^2\Big] = Dd\sigma^2\mathrm{Var}\big(\Delta c_k\big) \tag{8}$$

where $\Delta\boldsymbol{c} \overset{\text{def}}{=} \boldsymbol{c}' - \boldsymbol{c}$. The vectors $\boldsymbol{c}$ and $\boldsymbol{c}'$ are the current step's attention content before and after memory expansion, respectively, shown in Figure 2b, Appendix A. (We omit the time step in the notation for simplicity.) $W_{(c)}$ is the weight matrix connecting attention content to RNN states.

To prove the theorem, it remains to show that $\mathrm{Var}(\Delta c_k) \leq \sigma^2$. We do this by analyzing how attention is computed at each time step, and bounding each attention weight. For details, see the complete proof in Appendix A. □

In the theorem, we have an assumption $\sum_{j=N+1}^{N+M} \widetilde{\alpha}_{i,j} \leq \sum_{j=1}^{N} \widetilde{\alpha}_{i,j}$, requiring that the total attention to existing memory slots is larger than to the progressively added slots. This is fairly reasonable because: (1) During training, attention is trained in an *ad hoc* fashion to newly-added information, and thus some $\alpha_{i,j}$ for $1 \leq j \leq N$ might be learned so that it is larger than a random memory slot; and (2) For a new domain, we do not add a huge number of slots, and thus $\sum_{j=N+1}^{N+M} \widetilde{\alpha}_{i,j}$ will not dominate.

It is noted that our theorem is not to provide an explicit optimization/generalization bound for IDA, but shows that expanding memory is more stable than expanding hidden states. This is particularly important at the beginning steps of IDA, as the progressively growing parameters are randomly initialized and are basically noise. Although our theoretical analysis uses a restricted setting (i.e., vanilla RNN transition and linear activation), it provides the key insight that our approach is appropriate for IDA.

## 4 EXPERIMENTS

In this section, we evaluate our approach on an NLP classification task. In particular, we choose the multi-genre natural language inference (MultiNLI), due to its large number of samples in various domains. The task is to determine the relationship between two sentences among target labels: *entailment*, *contradiction*, and *neutral*. In Appendix C, we conduct supplementary experiments on text generation with our memory-augmented RNN for IDA.

**Dataset and Setup.** The MultiNLI corpus (Williams et al., 2018) is particularly suitable for IDA, as it contains training samples for 5 genres: `Slate`, `Fiction (Fic)`, `Telephone (Tel)`, `Government (Gov)`, and `Travel`. In total, we have 390k training samples. The corpus also contains held-out (non-training) labeled data in these domains. We split it into two parts for validation and test.[4]

---

[4] MultiNLI also contains 5 genres without training samples, namely, `9/11`, `Face-to-face`, `Letters`, `OUP`, and `Verbatim`. We ignore these genres, because we focus on incremental domain adaptation instead of

| Training domains | Performance on | | | | |
|---|---|---|---|---|---|
| | Fic | Gov | Slate | Tel | Travel |
| Fic | 65.41 | 58.87 | 55.83 | 61.39 | 57.35 |
| Fic → Gov | 67.55 | 70.82 | 61.04 | 65.07 | 61.90 |
| Fic → Gov → Slate | 67.04 | 71.55 | 63.29 | 64.66 | 63.53 |
| Fic → Gov → Slate → Tel | 68.46 | 71.10 | 63.39 | **71.60** | 61.50 |
| Fic → Gov → Slate → Tel → Travel | **69.36** | **72.47** | **63.96** | 69.74 | **68.39** |

Table 3: Dynamics of the progressive memory network for IDA with 5 domains. Upper-triangular values in gray are out-of-domain (zero-shot) performance.

| Group | Setting | Fic | Gov | Slate | Tel | Travel |
|---|---|---|---|---|---|---|
| Non-IDA | In-domain training | $65.41^{\Downarrow}$ | $67.01^{\Downarrow}$ | $59.30^{\Downarrow}$ | $67.20^{\Downarrow}$ | $64.70^{\Downarrow}$ |
| | Fic + Gov + Slate + Tel + Travel (multi-task) | $\mathbf{70.60}^{\uparrow}$ | **73.30** | 63.80 | 69.15 | $67.07^{\downarrow}$ |
| IDA | Fic → Gov → Slate → Tel → Travel (F+V) | $67.24^{\downarrow}$ | $70.82^{\Downarrow}$ | $62.41^{\downarrow}$ | $67.62^{\downarrow}$ | **68.39** |
| | Fic → Gov → Slate → Tel → Travel (F+V+M) | _69.36_ | _72.47_ | **63.96** | **69.74** | 68.39 |
| | Fic → Gov → Slate → Tel → Travel (EWC) | $67.12^{\Downarrow}$ | $68.71^{\Downarrow}$ | $59.90^{\Downarrow}$ | $66.09^{\Downarrow}$ | $65.70^{\Downarrow}$ |
| | Fic → Gov → Slate → Tel → Travel (Progressive) | $65.22^{\Downarrow}$ | $67.87^{\Downarrow}$ | $61.13^{\Downarrow}$ | $66.96^{\Downarrow}$ | 67.90 |

Table 4: Comparing our approach with variants and previous work in the multi-domain setting. In this experiment, we use the memory-augmented RNN as the neural architecture. Italics represent best results in the IDA group. $\uparrow, \downarrow$: $p < 0.05$ and $\Uparrow, \Downarrow$: $p < 0.01$ (compared with F+V+M).

The first row in Table 1 shows the size of the training set in each domain. As seen, the corpus is mostly balanced across domains, although `Tel` has slightly more examples.

For the base model, we train a bi-directional LSTM (BiLSTM). The details of network architecture, training, and hyper-parameter tuning are given in Appendix B. We see in Table 1 that we achieve similar performance to Yu et al. (2018). Furthermore, our BiLSTM achieves an accuracy of 68.37 on the official MultiNLI test set,[5] which is better than 67.51 reported in the original MultiNLI paper (Williams et al., 2018) using BiLSTM. This shows that our implementation and tuning are fair for the basic BiLSTM, and that our model is ready for the study of IDA.

**Transfer between Two Domains.** We would like to compare our approach with a large number of baselines and variants. Thus, we randomly choose two domains as a testbed: `Fic` as the source domain and `Gov` as the target domain. We show results in Table 2.

First, we analyze the performance of RNN and the memory-augmented RNN in the non-transfer setting (Lines 1–2 vs. Lines 3–4). As seen, the memory-augmented RNN achieves slightly better but generally similar performance, compared with RNN (both with LSTM units). This shows that, in the non-transfer setting, the memory bank does not help the RNN much. However, this later confirms that the performance improvement is indeed due to our IDA technique, instead of simply a better neural architecture.

We then apply two straightforward methods of domain adaptation: multi-task learning (Line 5) and fine-tuning (Line 6). Multi-task learning jointly optimizes source and target objectives, denoted by "S+T." On the other hand, the fine-tuning approach trains the model on the source first, and then fine-tunes on the target. In our experiments, these two methods perform similarly on the target domain, which is consistent with Mou et al. (2016). On the source domain, fine-tuning performs significantly worse than multi-task learning, as it suffers from the catastrophic forgetting problem. We notice that, in terms of source performance, the fine-tuning approach (Line 6) is slightly better than trained on the source domain only (Line 3). This is probably because our domains are somewhat correlated as opposed to Kirkpatrick et al. (2017), and thus training with more data on target slightly improves the performance on source. However, fine-tuning does achieve the worst performance on source compared with other domain adaptation approaches (among Lines 5–8). Thus, we nevertheless use the terminology "catastrophic forgetting," and our research goal is still to improve IDA performance.

The main results of our approach are Lines 7 and 8. We apply the proposed progressive memory network to IDA and we fine-tune all weights. We see that on both source and target domains,

---

zero-shot learning. Also, the labels for the official test set of MultiNLI are not publicly available, and therefore we cannot use it to evaluate performance on individual domains. Our split of the held-out set for validation and test applies to all competing methods, and thus is a fair setting.

[5]Evaluation on the official MultiNLI test set requires submission to Kaggle.

our approach outperforms the fine-tuning method alone where the memory size is not increased (comparing Lines 7 and 6). This verifies our conjecture that, if the model capacity is increased, the new domain results in less overriding of the learned knowledge in the neural network. Our proposed approach is also "orthogonal" to the expansion of the vocabulary size, where target-specific words are randomly initialized and learned on the target domain. As seen, this combines well with our memory expansion and yields the best performance on both source and target (Line 8).

We now compare an alternative way of increasing model capacity, i.e., expanding hidden states (Lines 9 and 10). For fair comparison, we ensure that the total number of model parameters after memory expansion is equal to the number of model parameters after hidden state expansion. We see that the performance of hidden state expansion is poor especially on the source domain, even if we fine-tune all parameters. This experiment provides empirical evidence to our theorem that expanding memory is more robust than expanding hidden states.

We also compare the results with previous work on IDA. We re-implement[6] EWC (Kirkpatrick et al., 2017). It does not achieve satisfactory results in our application. We investigate other published papers using the same method and find inconsistent results: EWC works well in some applications (Zenke et al., 2017; Lee et al., 2017) but performs poorly on others (Yoon et al., 2018; Wu et al., 2018). Wen & Itti (2018) even report near random performance with EWC. We also re-implement the progressive neural network (Rusu et al., 2016). We use the target predictor to do inference for both source and target domains. Progressive neural network yields low performance, particularly on source, probably because the predictor is trained with only the target domain.

We measure the statistical significance of the results against Line 8 with the one-tailed Wilcoxon's signed-rank test (Wilcoxon, 1945), by bootstrapping a subset of 200 samples for 10 times with replacement. The test shows our approach is significantly better than others, on both source and target domains.

**IDA with All Domains.** Having analyzed our approach, baselines, and variants on two domains in detail, we are now ready to test the performance of IDA with multiple domains, namely, `Fic`, `Gov`, `Slate`, `Tel`, and `Travel`. In this experiment, we assume these domains come one after another, and our goal is to achieve high performance on all domains.

Table 3 shows the dynamics of IDA with our progressive memory network. Comparing the upper-triangular values (in gray, showing out-of-domain performance) with diagonal values, we see that our approach can be quickly adapted to the new domain in an incremental fashion. Comparing lower-triangular values with the diagonal, we see that our approach does not suffer from the catastrophic forgetting problem as the performance of previous domains is gradually increasing if trained with more domains. After all data are observed, our model achieves the best performance in most domains (last row in Table 3), despite the incremental nature of our approach.

We now compare our approach with other baselines and variants in the multi-domain setting, shown in Table 4. Due to the large number of settings, we only choose a selected subset of variants from Table 2 for the comparison.

As seen, our approach of progressively growing memory achieves the same performance as fine-tuning on the last domain (both with vocabulary expansion), but for all previous 4 domains, we achieve significantly better performance than fine-tuning. Our model is comparable to multi-task learning on all domains. It should also be mentioned that multi-task learning requires training the model when data from all domains are available simultaneously. It is not an *incremental* approach for domain adaptation, and thus cannot be applied to the scenarios introduced in Section 1. We include this setting mainly because we are curious about the performance of non-incremental domain adaptation.

We also compare with previous methods for IDA in Table 4. Our method outperforms EWC and the progressive neural network in all domains; the results are consistent with Table 2.

---

[6]Implementation based on `https://github.com/ariseff/overcoming-catastrophic`

## 5 CONCLUSION

In this paper, we propose a progressive memory network for incremental domain adaptation (IDA). We augment an RNN with an attention-based memory bank. During IDA, we add new slots to the memory bank and tune all parameters by back-propagation. Empirically, the progressive memory network does not suffer from the catastrophic forgetting problem as in naïve fine-tuning. Our intuition is that the new memory slots increase the neural network's model capacity, and thus, the new knowledge causes significantly less overriding of the existing network. Compared with expanding hidden states, our progressive memory bank provides a more robust way of increasing model capacity, shown by both a theorem and experiments. We also outperform previous work for IDA, including elastic weight consolidation (EWC) and the original progressive neural network.

## ACKNOWLEDGMENTS

This work was funded by Huawei Technologies, Hong Kong. Lili Mou is supported by the Amii Fellow Program, and the CIFAR AI Chair Program.

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

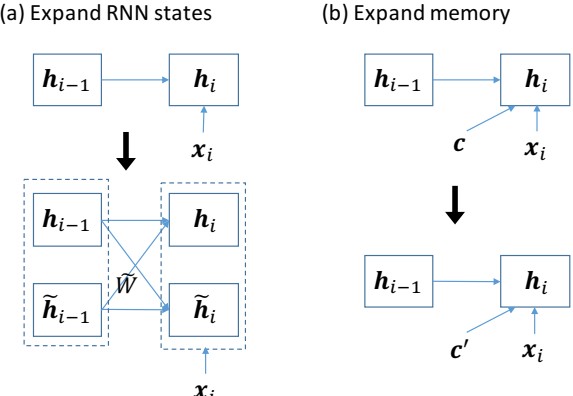

Figure 2: Hidden state expansion vs. memory expansion at step $t$.

## A  PROOF OF THEOREM 1

**Theorem 1.** *Let RNN have vanilla transition with the linear activation function, and let the RNN state at the last step $\boldsymbol{h}_{i-1}$ be fixed. For a particular data point, if the memory attention satisfies $\sum_{j=N+1}^{N+M} \widetilde{\alpha}_{i,j} \leq \sum_{j=1}^{N} \widetilde{\alpha}_{i,j}$, then memory expansion yields a lower expected mean squared difference in $\boldsymbol{h}_i$ than RNN state expansion. That is,*

$$\mathbb{E}\left[\|\boldsymbol{h}_i^{(\mathrm{m})} - \boldsymbol{h}_i\|^2\right] \leq \mathbb{E}\left[\|\boldsymbol{h}_i^{(\mathrm{s})} - \boldsymbol{h}_i\|^2\right] \tag{9}$$

*where $\boldsymbol{h}_i^{(\mathrm{m})}$ refers to the hidden states if the memory is expanded. $\boldsymbol{h}_i^{(\mathrm{s})}$ refers to the original dimensions of the RNN states, if we expand the size of RNN states themselves. Here, we compute the expectation by assuming weights and hidden states are iid from a zero-mean Gaussian distribution (with variance $\sigma^2$).*

*Proof:* Let $\boldsymbol{h}_{i-1}$ be the hidden state of the last step. We focus on one step of transition and assume that $\boldsymbol{h}_{i-1}$ is the same when the model capacity is increased. We consider a simplified case where the RNN has vanilla transition with the linear activation function. We measure the effect of model expansion quantitatively by the expected square difference on $\boldsymbol{h}_i$ before and after model expansion.

Suppose the original hidden state $\boldsymbol{h}_i$ is $D$-dimensional. We assume each memory slot is $d$-dimensional, and that the additional RNN units when expanding the hidden state are also $d$-dimensional. We further assume each variable in the expanded memory and expanded weights ($\widetilde{W}$ in Figure 2) are iid with zero mean and variance $\sigma^2$. This assumption is reasonable as it enables a fair comparison of expanding memory and expanding hidden states. Finally, we assume every variable in the learned memory slots, i.e., $m_{jk}$, follows the same distribution (zero mean, variance $\sigma^2$).

We compute how the original dimensions in the hidden state are changed if we expand RNN. We denote the expanded hidden states by $\widetilde{\boldsymbol{h}}_{i-1}$ and $\widetilde{\boldsymbol{h}}_i$ for the two time steps. We denote the weights connecting from $\widetilde{\boldsymbol{h}}_{i-1}$ to $\boldsymbol{h}_i$ by $\widetilde{W} \in \mathbb{R}^{D \times d}$. We focus on the original $D$-dimensional space, denoted as $\boldsymbol{h}_i^{(s)}$. The connection is shown in Figure 2a. We have

$$\mathbb{E}\left[\|\boldsymbol{h}_i^{(\mathrm{s})} - \boldsymbol{h}_i\|^2\right]$$

$$= \mathbb{E}\left[\|\widetilde{W} \cdot \widetilde{\boldsymbol{h}}_{i-1}\|^2\right] \tag{10}$$

$$= \mathbb{E}\left[\sum_{j=1}^{D}\left(\widetilde{\boldsymbol{w}}_j^{\top} \widetilde{\boldsymbol{h}}_{i-1}\right)^2\right] \tag{11}$$

$$= \sum_{j=1}^{D} \mathbb{E}\left[\left(\widetilde{\boldsymbol{w}}_j^{\top} \widetilde{\boldsymbol{h}}_{i-1}\right)^2\right] \tag{12}$$

$$= \sum_{j=1}^{D} \mathbb{E}\left[\left(\sum_{k=1}^{d} \widetilde{w}_{jk}\widetilde{h}_{i-1}[k]\right)^2\right] \tag{13}$$

$$= \sum_{j=1}^{D}\sum_{k=1}^{d} \mathbb{E}\left[\left(\widetilde{w}_{jk}\widetilde{h}_{i-1}[k]\right)^2\right] \tag{14}$$

$$= \sum_{j=1}^{D}\sum_{k=1}^{d} \mathbb{E}\left[\left(\widetilde{w}_{jk}\right)^2\right]\mathbb{E}\left[\left(\widetilde{h}_{i-1}[k]\right)^2\right] \tag{15}$$

$$= D \cdot d \cdot \mathrm{Var}\left(w\right) \cdot \mathrm{Var}(h) \tag{16}$$

$$= Dd\sigma^2\sigma^2 \tag{17}$$

where (14) is due to the independence and zero-mean assumptions of every element in $\widetilde{W}$ and $\boldsymbol{h}_{i-1}$. (15) is due to the independence assumption between $\widetilde{W}$ and $\boldsymbol{h}_{i-1}$.

Next, we compute the effect of expanding memory slots. Notice that $\|\boldsymbol{h}_i^{(\mathrm{m})} - \boldsymbol{h}_i\| = W_{(\mathrm{c})}\Delta\boldsymbol{c}$. Here, $\boldsymbol{h}_i^{(\mathrm{m})}$ is the RNN hidden state after memory expansion. $\Delta\boldsymbol{c} \stackrel{\text{def}}{=} \boldsymbol{c}' - \boldsymbol{c}$, where $\boldsymbol{c}$ and $\boldsymbol{c}'$ are the attention content vectors before and after memory expansion, respectively, at the current time step.[7] $W_{(\mathrm{c})}$ is the weight matrix connecting attention content to RNN states. The connection is shown in Figure 2b. Reusing the result of (16), we immediately obtain

$$\mathbb{E}\left[\|\boldsymbol{h}_i^{(\mathrm{m})} - \boldsymbol{h}_i\|^2\right] \tag{18}$$

$$= \mathbb{E}\left[\|W_{(c)}\Delta\boldsymbol{c}\|^2\right] \tag{19}$$

$$= Dd\sigma^2\mathrm{Var}\left(\Delta c_k\right) \tag{20}$$

where $\Delta c_k$ is an element of the vector $\Delta\boldsymbol{c}$.

To prove Equation (2), it remains to show that $\mathrm{Var}(\Delta c_k) \leq \sigma^2$. We now analyze how attention is computed.

Let $\widetilde{\alpha}_1, \cdots, \widetilde{\alpha}_{N+M}$ be the unnormalized attention weights over the $N+M$ memory slots. We notice that $\widetilde{\alpha}_1, \cdots, \widetilde{\alpha}_N$ remain the same after memory expansion. Then, the original attention probability is given by $\alpha_j = \widetilde{\alpha}_j/(\widetilde{\alpha}_1 + \cdots + \widetilde{\alpha}_N)$ for $j = 1, \cdots, N$. After memory expansion, the attention probability becomes $\alpha'_j = \widetilde{\alpha}_j/(\widetilde{\alpha}_1 + \cdots + \widetilde{\alpha}_{N+M})$, illustrated in Figure 3. We have

$$\Delta\boldsymbol{c} = \boldsymbol{c}' - \boldsymbol{c} \tag{21}$$

$$= \sum_{j=1}^{N}(\alpha'_j - \alpha_j)\boldsymbol{m}_j + \sum_{j=N+1}^{N+M}\alpha'_j\boldsymbol{m}_j \tag{22}$$

$$= \sum_{j=1}^{N}\left(\frac{\widetilde{\alpha}_j}{\widetilde{\alpha}_1 + \cdots + \widetilde{\alpha}_{N+M}} - \frac{\widetilde{\alpha}_j}{\widetilde{\alpha}_1 + \cdots + \widetilde{\alpha}_N}\right)\boldsymbol{m}_j + \sum_{j=N+1}^{N+M}\left(\frac{\widetilde{\alpha}_j}{\widetilde{\alpha}_1 + \cdots + \widetilde{\alpha}_{N+M}}\right)\boldsymbol{m}_j$$

$$= \sum_{j=1}^{N}\left(\frac{-\widetilde{\alpha}_j\frac{\widetilde{\alpha}_{N+1} + \cdots + \widetilde{\alpha}_{N+M}}{\widetilde{\alpha}_1 + \cdots + \widetilde{\alpha}_N}}{\widetilde{\alpha}_1 + \cdots + \widetilde{\alpha}_{N+M}}\right)\boldsymbol{m}_j + \sum_{j=N+1}^{N+M}\left(\frac{\widetilde{\alpha}_j}{\widetilde{\alpha}_1 + \cdots + \widetilde{\alpha}_{N+M}}\right)\boldsymbol{m}_j \tag{23}$$

$$\stackrel{\text{def}}{=} \sum_{j=1}^{N+M}\beta_j\boldsymbol{m}_j \tag{24}$$

where

$$\beta_j \stackrel{\text{def}}{=} \begin{cases} \dfrac{-\widetilde{\alpha}_j\frac{\widetilde{\alpha}_{N+1} + \cdots + \widetilde{\alpha}_{N+M}}{\widetilde{\alpha}_1 + \cdots + \widetilde{\alpha}_N}}{\widetilde{\alpha}_1 + \cdots + \widetilde{\alpha}_{N+M}}, & \text{if } 1 \leq j \leq N \\[4mm] \dfrac{\widetilde{\alpha}_j}{\widetilde{\alpha}_1 + \cdots + \widetilde{\alpha}_{N+M}}, & \text{if } N+1 \leq j \leq N + M \end{cases} \tag{25}$$

---

[7]We omit the time step in the notation for simplicity.

| Memory | Unnormalized measure | Original attn. prob. | Expanded attn. prob. |
|--------|---------------------|---------------------|---------------------|
| $m_1$ | $\tilde{\alpha}_1$ | $\alpha_1$ | $\alpha'_1$ |
| $m_2$ | $\tilde{\alpha}_2$ | $\alpha_2$ | $\alpha'_2$ |
| ... | ... | ... | ... |
| $m_N$ | $\tilde{\alpha}_N$ | $\alpha_N$ | $\alpha'_N$ |
| $m_{N+1}$ | $\tilde{\alpha}_{N+1}$ | | $\alpha'_{N+1}$ |
| ... | ... | | ... |
| $m_{N+M}$ | $\tilde{\alpha}_{N+M}$ | | $\alpha'_{N+M}$ |

Figure 3: Attention probabilities before and after memory expansion.

By our assumption of total attention $\sum_{j=N+1}^{N+M} \widetilde{\alpha}_j \le \sum_{j=1}^{N} \widetilde{\alpha}_j$, we have

$$|\beta_j| \le |\alpha'_j|, \quad \forall 1 \le j \le N+M \tag{26}$$

Then, we have

$$\mathrm{Var}(\Delta c_k) = \mathbb{E}[(c'_k - c_k)^2] \quad \forall 1 \le k \le d \tag{27}$$

$$= \frac{1}{d}\mathbb{E}[\|\boldsymbol{c}' - \boldsymbol{c}\|^2] \tag{28}$$

$$= \frac{1}{d}\mathbb{E}\left[\sum_{k=1}^{d}\left(\sum_{j=1}^{N+M} \beta_j m_{jk}\right)^2\right] \tag{29}$$

$$= \frac{1}{d}\sum_{k=1}^{d}\mathbb{E}\left[\left(\sum_{j=1}^{N+M} \beta_j m_{jk}\right)^2\right] \tag{30}$$

$$= \frac{1}{d}\sum_{k=1}^{d}\sum_{j=1}^{N+M}\mathbb{E}\left[(\beta_j m_{jk})^2\right] \tag{31}$$

$$= \frac{1}{d}\sum_{k=1}^{d}\sum_{j=1}^{N+M}\mathbb{E}[\beta_j^2]\mathbb{E}[m_{jk}^2] \tag{32}$$

$$= \frac{1}{d}\sum_{k=1}^{d}\sum_{j=1}^{N+M}\mathbb{E}[\beta_j^2]\sigma^2 \tag{33}$$

$$= \sigma^2\mathbb{E}\left[\sum_{j=1}^{N+M}\beta_j^2\right] \tag{34}$$

$$\le \sigma^2\mathbb{E}\left[\sum_{j=1}^{N+M}(\alpha'_j)^2\right] \tag{35}$$

$$\le \sigma^2 \tag{36}$$

Here, (31) is due to the assumption that $m_{jk}$ is independent and zero-mean, and (32) is due to the independence assumption between $\beta_j$ and $m_{jk}$. To obtain (36), we notice that $\sum_{j=1}^{N+M} \alpha'_j = 1$ with $0 \le \alpha'_j \le 1$ ($\forall 1 \le j \le N+M$). Thus, $\sum_{j=1}^{N+M}(\alpha'_j)^2 \le 1$, concluding our proof. $\qquad\square$

## B  HYPERPARAMETERS

We choose the base model and most of its settings by following the original MultiNLI paper (Williams et al., 2018): 300D RNN hidden states, 300D pretrained GloVe embeddings (Pennington et al., 2014) for initialization, batch size of 32, and the Adam optimizer for training. The initial

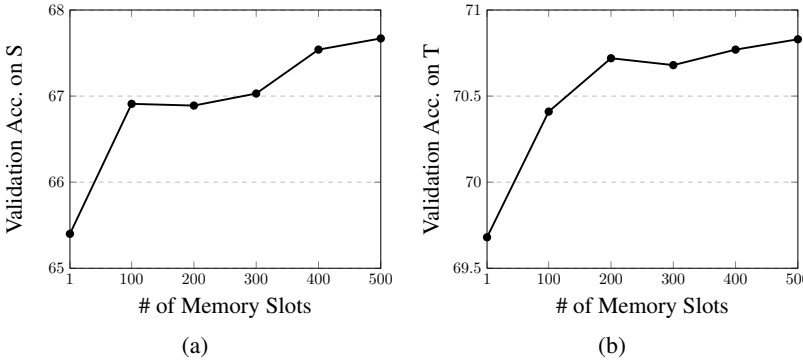

(a)                                                          (b)

Figure 4: Tuning the number of memory slots to be added per domain in the MultiNLI experiment. The two graphs show validation performance of our IDA model S→T (F+M+V).

learning rate for Adam is tuned over the set {0.3, 0.03, 0.003, 0.0003, 0.00003}. It is set to 0.0003 based on validation performance.

For the memory, we set each slot to be 300-dimensiooal, which is the same as the RNN and embedding size.

We tune the number of progressive memory slots in Figure 4, which shows the validation performance on the source (Fic) and target (Gov) domains. We see that the performance is close to fine-tuning alone if only one memory slot is added. It improves quickly between 1 and 200 slots, and tapers off around 500. We thus choose to add 500 slots for each domain.

## C  ADDITIONAL EXPERIMENT ON DIALOGUE GENERATION

We further evaluate our approach on the task of dialogue response generation. Given an input text sequence, the task is to generate an appropriate output text sequence as a response in human-computer dialogue. This supplementary experiment provides additional evidence of our approach in generation tasks.

**Datasets, Setup, and Metrics.** We use the Cornell Movie Dialogs Corpus (Danescu-Niculescu-Mizil & Lee, 2011) as the source. It contains ∼220k message-response pairs from movie transcripts. We use a 200k-10k-10k training-validation-test split.

For the target domain, we manually construct a very small dataset from the Ubuntu Dialogue Corpus (Lowe et al., 2015) to mimic the scenario where quick adaptation has to be done to a new domain with little training data. In particular, we choose a random subset of 15k message-response pairs, and use a 9k-3k-3k split.

The base model is a sequence-to-sequence (Seq2Seq) neural network (Sutskever et al., 2014) with attention from the decoder to the encoder. We use a single-layer RNN encoder and a single-layer RNN decoder, each containing 1024 cells following Sutskever et al. (2014). We use GRUs instead of LSTM units due to efficiency concerns. We have separate memory banks for the encoder and decoder, since they are essentially different RNNs. The source and target vocabularies are 27k and 10k, respectively. Each memory slot is 1024D, because the RNN states are 1024D in this experiment. For each domain, we progressively add 1024 slots; tuning the number of slots is done in a manner similar to the MultiNLI experiment. As before, we use Adam with an initial learning rate of 0.0003 and other default parameters.

Following previous work, we use BLEU-2 (Eric et al., 2017; Madotto et al., 2018) and average Word2Vec embedding similarity (W2V-Sim, Serban et al., 2017; Zhang et al., 2018a) as the evaluation metrics. BLEU-2 is the geometric mean of unigram and bigram word precision penalized by length, and correlates with human satisfaction to some extent (Liu et al., 2016). W2V-Sim is defined as the cosine similarity between the averaged Word2Vec embeddings of the model outputs

| # Line | Model | Trained on/by | BLEU-2 on | | W2V-Sim on | |
|---|---|---|---|---|---|---|
| | | | S | T | S | T |
| 1 | RNN | S | $2.842^⇑$ | $0.738^⇓$ | $0.480^⇓$ | $0.456^⇓$ |
| 2 | | T | $0.795^⇓$ | $1.265^⇓$ | $0.454^⇓$ | $0.480^⇓$ |
| 3 | RNN+ Mem | S | $\mathbf{3.074}^⇑$ | $0.712^⇓$ | $0.498^⇓$ | $0.471^⇓$ |
| 4 | | T | $0.920^⇓$ | $1.287^⇓$ | $0.462^⇓$ | $0.487^⇓$ |
| 5 | | S+T | $2.650^⇑$ | $0.889^⇓$ | $0.471^⇓$ | $0.462^⇓$ |
| 6 | RNN + Mem | S→T (F) | $1.210^⇓$ | $1.101^⇓$ | $0.509^⇓$ | $0.514^⇓$ |
| 7 | | S→T (F+M) | $1.435^⇓$ | $1.207^⇓$ | $\mathbf{0.526}$ | $0.522$ |
| 8 | | S→T (F+M+V) | $1.637$ | $\mathbf{1.652}$ | $0.522$ | $\mathbf{0.525}$ |
| 9 | | S→T (F+H) | $1.036^⇓$ | $1.606^↓$ | $0.503^⇓$ | $0.495^⇓$ |
| 10 | | S→T (F+H+V) | $1.257^⇓$ | $1.419^⇓$ | $0.504^⇓$ | $0.492^⇓$ |
| 11 | | S→T (EWC) | $1.397^⇓$ | $1.382^↓$ | $0.513^⇓$ | $0.514^⇓$ |
| 12 | | S→T (Progressive) | $1.299^⇓$ | $1.408^↓$ | $0.502^⇓$ | $0.503^⇓$ |

Table 5: Results on two-domain adaptation for dialogue response generation. F: Fine-tuning. V: Expanding vocabulary. H: Expanding RNN hidden states. M: Our proposed method of expanding memory. We also compare with EWC (Kirkpatrick et al., 2017) and progressive neural network (Rusu et al., 2016). $↑, ↓$: $p < 0.05$ and $⇑, ⇓$: $p < 0.01$ (compared with Line 8).

| Prompt | Response |
|---|---|
| i'm steve. what's your name? | my name is opera |
| i love you man | i love you too |
| so how's it going with your new roomie? | it should be fine |
| i didn't do anything wrong did i | we don't have much experience |
| i just got a messed up stomach | i know, you had to do some things |
| what are you trying to handicap me? | this is not really who i am |
| im scared | what do we do with it? |
| i kinda forgot the fix i tried a year ago | can you try the output ? |
| how would i remaster the ubuntu install | could you have a link both access to the network connection |
| my terminal is stuck | what version is it? |
| i am rebooting now | so your issue is a great work cd, sometimes i get some |
| i get an installation error | i am getting an expert for it |
| what does the kernel-package do? | you will need to get it to see if there is a setting |
| what directory should i be in? | you should be able to install the grub cd at the drive |

Table 6: Sample outputs of our IDA model S→T (F+M+V) from Table 5.

and the ground truths. Intuitively, BLEU measures hard word-level overlap between two sequences, whereas W2V-Sim measures soft similarity in a distributed semantic space.

**Results.** The results for dialogue response generation are shown in Table 5. We see that BLEU-2 and W2V similarity are not necessarily consistent. For example, the memory-augmented RNN trained solely on source achieves the best source BLEU-2, whereas the proposed progressive memory has the highest W2V cosine similarity on S. However, our model variants achieve the best performance on most metrics (Lines 7 and 8). Moreover, it consistently outperforms all other IDA approaches. Following the previous experiment, we conduct statistical comparison with Line 8. The test shows that our method is significantly better than the other IDA methods.

In general, the evaluation of dialogue systems is noisy due to the lack of appropriate metrics (Liu et al., 2016). Nevertheless, our experiment provides additional evidence of the effectiveness of our approach. It also highlights our model's viability for both classification and generation tasks.

**Case Study.** Table 6 shows sample outputs of our IDA model on test prompts from the Cornell Movie Corpus (source) and the Ubuntu Dialogue Corpus (target). We see that casual prompts from the movie domain result in casual responses, whereas Ubuntu queries result in Ubuntu-related responses. With the expansion of vocabulary, our model is able to learn new words like "grub"; with progressive memory, it learns Ubuntu jargon like "network connection." This shows evidence of the success of incremental domain adaptation.

