# OpenReview forum: "Progressive Memory Banks for Incremental Domain Adaptation"
_ICLR.cc/2020/Conference — Accept (Poster)_

### Official Review · AnonReviewer3 · 2019-10-23
**Official Blind Review #3**

**Rating:** 6

**Review:**

This paper proposes an extensible attention mechanism applied on the previous hidden state of an RNN and resulting in supplementary input for the next RNN step. For each added domain, new pairs of attentions key and values can be added to provide more capacity for the model. This method is applied in the context of incremental domain adaptation for NLP without the possibility of storing of old samples (episodic memory).

Pros:
- Extensive ablation study with the different possible combinations of methods
- Very interesting comparison between expanding the memory (i.e. attention) and expanding the hidden states. Using the attention results in better results for a same number of added parameters and the activations sizes stay the same even when the attention is extended with new pairs.
- Paper is well written/motivated

Weaknesses:
- MultiNLI seem to have too much correlation between tasks. It would have been better to be able to observe catastrophic forgetting for the source domain. In the appendix, the metrics have really strong disagreement so it is tough to judge for these two corpuses.
- When you give the numbers for multi-task learning, you should use your extended memory method to be fair with MT learning. I would just be interested to see it, just as a proper upper bound.

Otherwise, the paper proposes a novel method which works well in practice so I am leaning towards acceptance.

**Experience Assessment:**

I have published one or two papers in this area.

**Review Assessment: Checking Correctness Of Derivations And Theory:**

I assessed the sensibility of the derivations and theory.

**Review Assessment: Checking Correctness Of Experiments:**

I assessed the sensibility of the experiments.

**Review Assessment: Thoroughness In Paper Reading:**

I read the paper thoroughly.

---

### Official Review · AnonReviewer1 · 2019-10-25
**Official Blind Review #1**

**Rating:** 6

**Review:**

*** Summary

This work proposes to use an augmented RNN model to address the incremental domain adaptation problem. In particular, it designs the progressive memory bank approach which expands the memory capacity by adding parameters every time a new task comes in. The RNN retrieves knowledge from the memory bank via key-value attention. A proof in a highly simplified case is given in addition to empirical results showing that expanding the memory bank is better than expanding the RNN states.

*** Strengths

1. Section 3 is well-written. The methods and motivations are illustrated clearly.

2. Comprehensive experiments are conducted. Supportive results for the arguments presented in Section 3 are therefore demonstrated.


*** Weaknesses

1. Regarding Table 2., multiple runs of different sources and targets would be helpful to better understand the effectiveness of the proposed methods in the 2-domain set-up.

2. The choice of key-value memory bank is not intuitive. A comparison between this memory and the traditional attention can help demonstrate the validity of this choice.

**Experience Assessment:**

I have read many papers in this area.

**Review Assessment: Checking Correctness Of Derivations And Theory:**

I assessed the sensibility of the derivations and theory.

**Review Assessment: Checking Correctness Of Experiments:**

I carefully checked the experiments.

**Review Assessment: Thoroughness In Paper Reading:**

I read the paper thoroughly.

---

### Official Review · AnonReviewer4 · 2019-11-02
**Official Blind Review #4**

**Rating:** 6

**Review:**


###Summary###
This paper introduces incremental domain adaptation for natural language processing, assuming that each domain comes one after another and only the current domain can be accessed in the application scenario.  The basic framework of this paper is based on RNN but augmented with the directly parameterized memory bank.

The memory bank of this paper is a set of distributed, real-valued vectors capturing domain knowledge. When the model is adapted to the new domain, the model progressively increases the slots in the memory bank.

The paper evaluates the proposed approach on an NLP classification task, i.e. multi-genre natural language inference (MultiNLI). The dataset used in this paper includes 5 genres: Slate, Fiction, Telephone, Government and Travel.

In the experiments, the paper performs the dynamics of the progressive memory network for IDA as well as compares the proposed method with variants and previous work in the multi-domain setting.


### Novelty ###

This paper proposes incremental domain adaptation, which is inspired by Li & Hoiem's work. The setting assumes that each domain comes one after another and only one domain can get accessed. This setting is interesting as we will encounter this setting in the real application scenarios, i.e., the domain knowledge in the real domain is unpredictable. Thus, the problem setting provides some novelty. However, I am not sure whether assuming that we can only get access to one domain is reasonable as we can always save the data for the domain we have already seen.

From the perspective of the method, this paper incorporates the memory bank to the RNN, which is not new in the machine learning research area, but heuristic enough for the transfer learning community.



###Clarity###

Overall, the paper is well organized and logically clear. The proposed claims are well supported by the experiments and analysis. The images are well-presented and well-explained by the captions and the text.


###Pros###

1) The paper proposes an incremental domain adaptation scenario where one domain appears another and only the data from the current domain can be accessed, which is interesting and heuristic to the domain adaptation research community.
2) The paper is applicable to many practical scenarios since the data from the real-world application is typically from multiple domains and the data is from one domain at a time.
3) The paper is overall well-organized and well-written. The claims of the paper are verified by the experimental results.


###Cons###

1) The paper has a good motivation for the setting, however, one of the critical drawbacks of this paper is that the papers fail to compare with the state-of-the-art baselines. I understand that this paper has a new setting, but since the authors also compare the proposed method with the "multi-task" learning, it will be helpful to compare with state-of-the-art multi-task or multi-source baselines.
2) The experimental results provided in this paper are weak. In Table 4, we found that sometimes, the IDA method performs worse than the multi-task baselines.
3) The paper presents no ablation study or analysis of the experimental results. The effectiveness of the memory bank, fine-tuning/freezing learning parameters is unclear.

It will be also interesting to see how does the proposed method performs on large-scale visual datasets.



Based on the summary, cons, and pros, the current rating I am giving now is weak reject. I would like to discuss the final rating with other reviewers, ACs.
I am willing to improve my rating if the authors can address my following concerns. To improve the rating, the author should explain the following questions:
1). Why assuming that we can only get access to the data from one domain is a reasonable setting, since we can always save the data (or at least the statistics about the data) form the domains we have already observed.
2). Can the proposed approach generalize to visual domain adaptation, i.e. on the visual task instead of NLP task?
3). The drawbacks I mentioned in the paper Cons section.



#################### Updated Review  ###########
The authors have addressed most of my concerns I proposed in the initial review, thus I raise my score to 6 weak accept.

1) In the response to the review, the authors claim that from the data privacy's perspective, it's reasonable to assume that we can only get access to one dataset at one time, which makes sense.
2) I would be really interested in how the similar idea works on vision data, which could be a future work for this paper.

**Experience Assessment:**

I have published one or two papers in this area.

**Review Assessment: Checking Correctness Of Derivations And Theory:**

I assessed the sensibility of the derivations and theory.

**Review Assessment: Checking Correctness Of Experiments:**

I carefully checked the experiments.

**Review Assessment: Thoroughness In Paper Reading:**

I read the paper at least twice and used my best judgement in assessing the paper.

---

### Decision · Program_Chairs · 2019-12-19

**Decision:**

Accept (Poster)

**Comment:**

This paper introduces an RNN based approach to incremental domain adaptation in natural language processing, where the RNN is progressively augmented with the parameterized memory bank which is shown to be better than expanding the RNN states.

Reviewers and AC acknowledge that this paper is well written with interesting ideas and practical value. Domain adaptation in the incremental setting, where domains come in a streaming way with only the current one accessible, can find some realistic application scenarios. The proposed extensible attention mechanism is solid and works well on several NLP tasks. Several concerns were raised by the reviewers regarding the comparative and ablation studies, which were well resolved in the rebuttal. The authors are encouraged to generalize their approach to other application domains other than NLP to show the generality of their approach.

I recommend acceptance.